# Tree Shrew as an Emerging Small Animal Model for Human Viral Infection: A Recent Overview

**DOI:** 10.3390/v13081641

**Published:** 2021-08-18

**Authors:** Mohammad Enamul Hoque Kayesh, Takahiro Sanada, Michinori Kohara, Kyoko Tsukiyama-Kohara

**Affiliations:** 1Transboundary Animal Diseases Centre, Joint Faculty of Veterinary Medicine, Kagoshima University, Kagoshima 890-0065, Japan; mehkayesh@pstu.ac.bd; 2Department of Microbiology and Public Health, Faculty of Animal Science and Veterinary Medicine, Patuakhali Science and Technology University, Barishal 8210, Bangladesh; 3Department of Microbiology and Cell Biology, Tokyo Metropolitan Institute of Medical Science, Tokyo 156-8506, Japan; sanada-tk@igakuken.or.jp (T.S.); kohara-mc@igakuken.or.jp (M.K.)

**Keywords:** tree shrew, animal model, hepatitis virus, respiratory virus, arbovirus, infection

## Abstract

Viral infection is a global public health threat causing millions of deaths. A suitable small animal model is essential for viral pathogenesis and host response studies that could be used in antiviral and vaccine development. The tree shrew (*Tupaia belangeri* or *Tupaia belangeri chinenesis*), a squirrel-like non-primate small mammal in the Tupaiidae family, has been reported to be susceptible to important human viral pathogens, including hepatitis viruses (e.g., HBV, HCV), respiratory viruses (influenza viruses, SARS-CoV-2, human adenovirus B), arboviruses (Zika virus and dengue virus), and other viruses (e.g., herpes simplex virus, etc.). The pathogenesis of these viruses is not fully understood due to the lack of an economically feasible suitable small animal model mimicking natural infection of human diseases. The tree shrew model significantly contributes towards a better understanding of the infection and pathogenesis of these important human pathogens, highlighting its potential to be used as a viable viral infection model of human viruses. Therefore, in this review, we summarize updates regarding human viral infection in the tree shrew model, which highlights the potential of the tree shrew to be utilized for human viral infection and pathogenesis studies.

## 1. Introduction

Viral infection is a major threat to public health, causing millions of deaths worldwide. Animal models mimicking human diseases are invaluable tools to study viral infection and pathogenesis, as well as for the development of prophylactic and/or therapeutic interventions. In the medical community, the term “tree shrew” or “Tupaia” refers to *Tupaia belangeri* (northern tree shrew) or *T. belangeri chinensis* (Chinese tree shrew); it belongs to the order Scandentia, which splits into two families, Ptilocercidae and Tupaiidae [1]. Tree shrews belong to the Tupaiidae family and the genus Tupaia, which includes 16 species [2,3]. Taxonomically, *T. belangeri chinensis* is a subspecies of *T. belangeri*, but genetically they show differences [2]. In addition, high genetic diversity within *T. belangeri* is also observed [1]. Moreover, genetic variation was also noticed within *T. belangeri chinensis* [4]. Although the tree shrew shows potential as animal model, genetic instability should be carefully considered during evaluation of experimental results. Tree shrews are a squirrel-like non-primate mammal with a body weight ranging between 50 and 270 g. Tree shrews are widely distributed in Southeast Asia, including South and Southwest China. At the age of six to nine months old, tree shrews can give birth on average to four babies with a gestation period of 45 days [3]. Tree shrews are genetically closer to humans than to rodents [5,6]. The tree shrew has been extensively used as an animal model for biomedical research of different conditions, including myopia [7], depression [8,9], cancer [10,11,12], chemotherapy models [13], and non-alcoholic fatty liver disease [14], to name a few. Of note, the tree shrew has also been used for viral infections, and its use in human viral infections is increasing; thus, the tree shrew emerges as a potential small animal model in advancing the knowledge of viral pathogenesis and diseases [3,15]. At the present time, tree shrews have been used as models for several important human viral pathogens (Figure 1).

The tree shrew has unique characteristics that render it a potentially advantageous animal model, including small body size, short reproductive cycle and life span, easy handling, low maintenance cost, and genetic closeness to primates. Although the tree shrew appears as a promising animal model, its extensive use is limited by the availability of tree shrew-specific reagents, individual variability, etc. [16]. As the whole genome sequence and tree shrew database are available now, these limitations should gradually decrease and the use of the tree shrew should increase. Here, we provide an update on the tree shrew viral infection model for human-infecting viruses (Table 1), including HBV, HCV, influenza virus, SARS-CoV-2, HAdV, ZIKV, and DENV infection and pathogenesis, which will enhance our understanding.

## 2. Tree Shrew Model for Hepatitis B and Hepatitis C Virus

### 2.1. Hepatitis B Virus (HBV)

HBV is a major global public health problem causing chronic hepatitis, liver cirrhosis, and/or hepatocellular carcinoma (HCC) [37]. HBV has a very restricted host range, and previously the chimpanzee (*Pan troglodytes*) has been used as the only natural-infection model [38]; however, its use in experimental infection has become difficult due to high expense and ethical and animal welfare issues. An alternative HBV-permissive animal model is essential, and the tree shrew has been reported to be susceptible to HBV infection. However, the infection rate is very low, and the tree shrew transiently carries the virus [18,39]. However, with the use of tree-shrew adapted HBV virus, the limitation of low infection rate in the tree shrew model should be overcome [39]. HBV induces HCC in tree shrews which is synergistically enhanced with aflatoxin B1 exposure [40]. Including our group, several groups have been working to establish a tree shrew HBV infection model. The tree shrew has been modeled for both acute and chronic HBV infection [17,19,20,41,42]. In our previous study, we observed a varied replication efficiency due to HBV genotype differences, where HBV-A2 propagation was higher than that of HBV-C; however, the overall viral titer was very low [17]. However, the effect of high genetic variation of the animal on the replication efficiency of HBV genotypes cannot be ruled out. There are eight well-known HBV genotypes (A–H) [43], which are characterized by greater than 8% difference in the nucleotide sequence of the HBV genome [44], and the characterization of all the HBV genotypes in tree shrew is yet to be completed. In our study, we observed that interferon β (IFN-β) was significantly suppressed in chronic HBV infection at 31 weeks post infection in all HBV-infected tree shrews, suggesting the impairment of the interferon response in chronic infection, whereas in acute infection the IFN-β response was variable [17]. Tree shrews show a tendency of chronic infection when infected as neonates and similar liver histopathological changes are observed as in human infections [17,19,20]. Additionally, primary tupaia hepatocytes (PTHs) are widely used for HBV infection studies, which should provide further insights into HBV pathogenesis [45,46,47]. Of note, sodium taurocholate co-transporting polypeptide (NTCP), the cellular entry receptor for HBV, was recently identified using PTHs [45]. Recently, transplantation of tupaia primary hepatocytes into chimeric mice were found to be useful and resulted in efficient HBV infection [47]. It has been reported that tumor necrosis factor α (TNF-α) can suppress HBV replication in PTHs [48]. Thus, the tree shrew model of HBV infection has been an invaluable tool for HBV study.

### 2.2. Hepatitis C Virus (HCV)

HCV is a major global health problem; it infects approximately 130–170 million people worldwide and causes chronic liver disease, cirrhosis, and HCC [49]. Similar to HBV, HCV also has a narrow host range, with spreading replication reported exclusively in humans and chimpanzees. Lack of small animal models is a significant obstacle in the field of HCV research. However, in vitro and in vivo tree shrew models have been very useful in the investigation of many aspects of the HCV life cycle, including HCV replication, pathogenesis and antiviral interventions [22,50,51,52,53]. However, HCV-infected tree shrews showed intermittent viral propagation and low viral load [21,22,23,51], which is suggestive of restricted replication in the tree shrew compared to humans. Additionally, liver histopathological changes observed in tree shrews are comparable to human cases [21,22]. Notably, a previous study showed the development of an acute infection that led to chronicity in the tree shrew model which was characterized by the development of liver steatosis, cirrhotic nodules and tumorigenesis [21]. In our previous study we demonstrated the HCV-specific antibody response in the tree shrew, although the titers were found to fluctuate [22], suggesting the tree shrew might be useful in preclinical vaccine efficacy testing. Moreover, the innate immune response, such as toll-like receptors and cytokines, were found to be induced in HCV infection [22], suggesting the tree shrew could serve as a potential model for an immune response and pathogenesis study.

## 3. Tree Shrew Model for Respiratory Viruses

### 3.1. Influenza Virus

The use of the tree shrew in influenza research was first introduced in 2013. As demonstrated, the tree shrew supports the replication of human influenza viruses without prior adaptation, producing mild disease symptoms [24]. The entry receptors of both human influenza virus (SAα2,6 Gal) and avian influenza virus (SAα2,3 Gal) were found to be distributed in the tree shrews’ respiratory tract [24]. In another study, it has been demonstrated that avian influenza A (H9N2) virus infection in tree shrews and ferrets showed comparable pathogenicity [25]. There was induction of a variable cytokine response in respiratory tissues at different days post-infection (dpi), including TNF-α, IFN-β, IL-4, IL-6, CXCL8 (IL-8), IL-10, IL-13, CXCL10 (IP-10), CCL5 (RANTES), CXCL9 (MIG), and CCL2 (MCP-1) [25]. In our previous study, we observed that multisite inoculation of H5N1 and H7N9 avian influenza virus with 1.0 × 10^6^ plaque-forming units (PFU) produced severe diffuse pneumonia and focal pneumonia, respectively [26]. The severity of pneumonia was found to be correlated with proinflammatory cytokine transcript levels. Moreover, H5N1 also induced fever and body weight loss; however, no effect on body weight change by H7N9 was observed. We also observed antibody induction in tree shrews infected with H5N1 and H7N9 [26], suggesting that the tree shrew could be used for evaluating antigen-specific antibody response and for vaccine efficacy studies [54]. A recent study evaluated the infectivity and transmissibility of three unadapted different strains of influenza A virus, including pandemic H1N1 (A/Sichuan/1/2009, pdmH1N1), avian-origin H5N1 (A/Chicken/Gansu/2/2012, H5N1) and early human-origin H7N9 (A/Suzhou/SZ19/2014, H7N9), in tree shrews, and showed efficient replication and subclinical symptoms in tree shrews [27]. A strong humoral immune response was induced in virus-inoculated tree shrews, which was protective in a challenge with a homologous virus [27]. In a recent study, it has been demonstrated that intranasal inoculation of unadapted human influenza B (V0215 and Y12) with 1.0 × 10^6^ TCID50 could infect tree shrews and produce milder nasal secretions and weight loss with no significant respiratory symptoms. However, an increase in cytokine levels of IL-6, IL-10, IP-10, TNF-α and TGF-β mRNA in tissues were observed [28]. These results indicated the suitability of tree shrews in studies of influenza viruses, particularly considering their relatively small size, evolutionary status as closer to primates and lower cost compared to ferrets. Findings obtained from different studies with influenza virus infection in tree shrews suggest that the tree shrew is a viable animal model for influenza research.

### 3.2. Severe Acute Respiratory Syndrome Coronavirus 2 (SARS-CoV-2)

SARS-CoV-2 is the causal agent of the ongoing COVID-19 pandemic, which emerged in 2019 in Wuhan, China, spread globally and resulted millions of deaths [55,56]. SARS-CoV-2 belongs to the family Coronaviridae and the genus Betacoronavirus, and SARS-CoV-2 is the third CoV that involved spillover of an animal coronavirus to humans [57]. The pathogenesis of SARS-CoV-2 is complex and poorly understood [58]. Different animal models have been investigated for advancing the understanding of SARS-CoV-2 pathogenesis and host response, including mouse models, Syrian hamster model, ferret models, non-human primate models, and additional animal models including mink, cat, dog, pig, chicken, duck, and fruit bats [59]. Moreover, a suitable animal model is crucial for the development of targeted preventions and treatments. A recent study showed that tree shrews could be infected with SARS-CoV-2 strain 107 (1 × 10^7^ TCID50) using multiple routes, such as oral, intranasal, and ocular routes. Viral load peaked at 3 dpi in lung tissues, and histopathology revealed hemorrhage in lung tissues of infected animals [30]. However, this study found old tree shrews (5–6-years-old) were less susceptible to SARS-CoV-2 infection compared to adult tree shrews (1-year-old), which is not compatible with cases as reported in humans, where older humans are more susceptible to SARS-CoV-2 infection than younger humans [60]. Another study reported increased body temperature in SARS-CoV-2-inoculated tree shrews, particularly in female animals; however, no clinical signs were observed [29], indicating the inconsistency of results in the study of SARS-CoV-2 infection in the tree shrew model, which could be due to its high genetic variation. However, the possibility of the tree shrew as an asymptomatic carrier or reservoir cannot be ruled out, which requires future investigation.

### 3.3. Human Adenoviruses (HAdVs)

HAdVs are non-enveloped, double-stranded DNA viruses with a genome of 36 kb belonging to the Adenoviridae family [61]. At the present time, seven HAdV species (A–G) and over 100 types (http://hadvwg.gmu.edu/; accessed on 30 May 2021) have been reported, based on genome sequencing, and more types remain to be identified [62]. HAdV infection often causes mild diseases; however, life-threatening respiratory disease may occur. Moreover, based on the infecting HAdV genotype, the clinical manifestation largely varies [63], and pneumonia has been reported as one of the more serious types of HAdV-induced respiratory disease in children [64]. A suitable animal model is essential for studying pathogenicity, promoting antiviral and vaccine development, and conducting preclinical efficacy testing of oncolytic adenoviral vectors. Moreover, a permissive animal model for HAdV-B infection is still lacking [31], which is crucial for a proper understanding of pathogenesis and host response. A recent study by Li et al. showed that tree shrews demonstrate susceptibility to HAdV-B infection. They showed that intranasally inoculated HAdV-55 efficiently replicates and produces severe interstitial pneumonia in tree shrews [31]. However, a pre-existing neutralizing antibody (nAb) was found in 6 tree shrews out of 18 against HAdV-3, and 1 had a preexisting nAb against HAdV-14, suggesting the occurrence of natural infection of HAdV in tree shrews. Notably, AdVs are wide spread and infect all five taxa of vertebrates. It has been reported that Tupaia adenovirus (TAV) infects the members of Tupaiidae family, and TAV has a genome of 33.5 kb [65]. However, there were no pre-existing nAbs against HAdV-55 in those tree shrews [31]. The overall findings of the study demonstrated the tree shrew as a permissive animal model for studying HAdV-B infection which may be useful for antiviral testing.

## 4. Tree Shrew Model for Arboviruses

### 4.1. Zika Virus (ZIKV)

Emerging and reemerging pathogens constitute a major public health threat globally [66]. ZIKV is a mosquito-borne flavivirus in the Flaviviridae family containing a positive-sense, single-stranded RNA genome of approximately 10.7 kb in size. ZIKV contains two major geographically different lineages, the Asian and African lineages [67]. Recently, ZIKV has spread worldwide and the WHO declared ZIKV fever as a public health emergency [68]. ZIKV has been associated with fetal microcephaly and serious neurological complications in adults [69]. A proper understanding of ZIKV pathogenesis and host-virus interaction is crucial for antiviral and vaccine development, for which a suitable small animal model is highly essential. There has been much progress in understanding of the ZIKV pathogenesis and host response using several animal models, including mice and non-human primates [70]; however, each model has its limitations. Therefore, an alternative animal model is required. Zhang et al. showed that tree shrews subcutaneously inoculated with ZIKV strain GZ01 (10^6^ PFU/animal) produced transient viremia, cutaneous inflammation and notable dermatological manifestations of skin rashes, as commonly observed in human patients [32]. However, there was no induction of fever or other neurological and behavioral abnormalities. Notably, ZIKV infection in adult tree shrews induced an antibody response that prevented reinfection with a homologous virus [32], suggesting the tree shrew model could be used for vaccine candidate screening. During neonatal infection of 1-day-old tree shrews through the intracerebral route with 10^5^ PFU of ZIKV, despite the detection of high levels of ZIKV RNA in the brain at 3 and 6 dpi and substantial pathological changes of neuron destruction, no signs of microcephaly in the ZIKV-infected tree shrews was observed [32]. In another study, in vitro tissue tropism of ZIKV infection in the tree shrew has been demonstrated, and the obtained findings indicated the susceptibility of the tree shrew primary cells from the kidney, lung, liver, skin and aorta to ZIKV infection [71]. Moreover, the antiviral innate immune response was induced, as observed by the induction of different cytokines, including IL-6, IL-8, TNF-α, IFN-β, CXCL9, and MX1 [71]. Although further studies are required, the above findings highlight the potential of the tree shrew for ZIKV infection and pathogenesis studies.

### 4.2. Dengue Virus (DENV)

DENV is the causal agent of dengue, a rapidly spreading mosquito-borne viral disease of tropical and sub-tropical areas. DENV belongs to the family Flaviviridae and the genus Flavivirus, and there are four DENV serotypes, DENV-1, DENV-2, DENV-3, and DENV-4 [72]. A suitable small animal model mimicking dengue in humans is still lacking. Previously, we showed the susceptibility of tupaia fibroblast cells to DENV infection with all four serotypes, where an upregulation of TLR8 and IL-8 mRNA expression was also observed in DENV-infected tupaia cells compared to uninfected cells [73]. Recently, the tree shrew animal model for dengue has been described by Jiang et al. as a novel model, and intravenous or subcutaneous inoculation of DENV-2 and DENV-3 in tree shrews showed some characteristics of dengue in humans [16]. As reported, there was an increased body temperature in DENV-infected tree shrews compared to uninfected animals. Moreover, proliferation of DENV and pathological changes in the brain were observed; however, viremia was very low [16]. Notably, no manifestation of severe dengue was reported. To extend the utility of the tree shrew dengue model, further research is required, including tree shrew-adapted DENV strain development, investigation of infection and pathogenesis with other serotypes, and suitability for studying the antibody-dependent enhancement (ADE) effect in the tree shrew model. Moreover, the susceptibility of tree shrews to other members of the flavivirus genus such as West Nile virus and Japanese encephalitis virus could be investigated, which should enhance the characterization of these viral infections and their pathogenesis.

## 5. Tree Shrew Model for Other Human Viruses

### 5.1. Herpes Simplex Virus (HSV)

HSV primarily infects human populations and causes a contagious infection affecting approximately 60–95% of adults globally. The HSV belongs to the family Herpesviridae, and has 2 types, HSV type 1 (HSV-1) and HSV type 2 (HSV-2) [74]. HSV-1 mainly infects the mouth, pharynx, face, eye, and central nervous system (CNS), while HSV-2 primarily infects the anogenital region [75,76]. An earlier report showed experimental infection of HSV-1 and HSV-2 (25–10^5^ PFU/animal) in juvenile (28–45 days old) and adult (150 days old) tree shrews when inoculated intravenously, intraperitoneally, or subcutaneously. This resulted in deaths in juvenile animals and acute illness in adults [36]. Natural infection of HSV-2 in tree shrews (1/272) was reported previously, as confirmed by the presence of anti-HSV-2 IgG [77]. A recent study reported that HSV-1 infection in tree shrew trigeminal ganglia (TG) through ocular inoculation has important similar features to human infection regarding expression patterns of key HSV-1 viral genes, including LAT and ICP0 during latency, that differs significantly from mice. Both the LAT and ICP0 regions were robustly transcribed in tree shrews and human TG during latency; however, only the LAT region was transcribed in mouse TG. Additionally, spontaneous reactivation of HSV-1 infection from latency in tree shrews has been reported [33], providing an alternative tool to study HSV-1 infection and pathogenesis. A previous study reported that HSV-1 can latently infect the CNS of the tree shrew [35]. Notably, the eye structures of tree shrew are similar to humans, and a recent study demonstrated that ocular inoculation of HSV-1 causing infection in tree shrew corneas resulted in inflammation of the cornea, known as herpes simplex virus keratitis (HSK), as observed in humans [34], suggesting that the tree shrew could be a useful model for studying HSK. Overall, HSV-1-related disease manifestations as observed in humans, such as latent infections and reactivations, are also observed in tree shrews, which highlights the suitability of the tree shrew model of HSV-1 infection.

### 5.2. Human Immunodeficiency Virus Type 1 (HIV-1)

A recent study characterized the ability of tree shrew cells to support HIV-1 in vitro. This study showed that the exogenous expression of human CD4 and CCR5 362 molecules in tree shrew cells support HIV-1 entry and replication and showed that the tree shrew CXCR4 serves as a functional co-receptor [78]. The tree shrew cytidine deaminases of the APOBEC3 family have been reported as the host restriction factor for HIV-1 in tree shrew cells, which exerted a strong inhibitory effect on HIV-1 replication [78]. Therefore, the findings of the study discovered APOBEC3 proteins as a barrier to HIV-1 infection; this information is critical for developing gene-edited tree shrew HIV-1-infected models. Previously, induction of hypermutations in the HIV-1 genome by APOBEC3 proteins was reported [79].

## 6. Conclusions

The development of animal models is essential for studying the pathogenesis of viral infections, as well as for evaluating the efficacy of drugs and vaccines. The use of the tree shrew model for viral infection is increasing due to its susceptibility to several important human viral pathogens. No animal models can be a perfect alternate to the human infection model. However, animal models mimicking human diseases are crucial for properly understanding the host response and pathogenesis, and at the same time, experiments should be conducted in such a way that the results obtained using animal models can be translated to humans. As the ethical and financial concerns limit the use of chimpanzees in hepatitis virus research, the tree shrew model could be developed. However, high genetic variation within this species may hinder a wide use of this animal in biomedical research. So far, no inbred line of tree shrews has been reported, which is required to be developed to reduce the individual variations found in wild tree shrews. Moreover, there is the challenge of the overall low viral titer in tree shrews and the scarcity of the research tools that should be overcome. However, the development of inbred tree shrews is underway [4], and a recent report of transgenic tree shrews has been released [80], which should further enhance the widespread use of tree shrews in biomedical research, including viral infections.

## Figures and Tables

**Figure 1 viruses-13-01641-f001:**
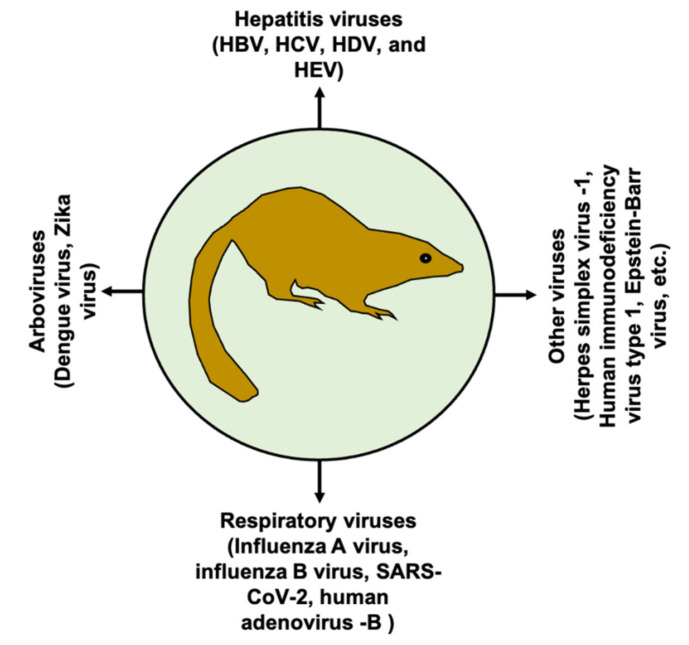
Tree shrew as a model of human viral infections. The tree shrew has been modeled for several important human viral pathogens as indicated in the figure.

**Table 1 viruses-13-01641-t001:** Human viral infections in the tree shrew model.

Virus Species	Serotype/Subtype/Genotypes/Strain	Inoculum Dose	Inoculation Route	Age of Animal	Viral Load or Titer (Plasma/Serum/Nasal Wash/Tissue)	Cytokine and Antibody Response	Clinical Signs/Strengths	Limitations	References
HBV	HBV genotype A2, C	1 × 10^6^–1 × 10^7^ GEs/animal	SC, IP	Newborn or adults (1 Y)	10–>10^3^ copies/mL serum	IFN-β response impaired in chronic infection, and variable in acute infection; TNF-α expression changes	Immunocompetent animal model; viremia develops; development of both acute and chronic infection; occurrence of histopathological changes in the liver; detection of intrahepatic HBV DNA	Individual variations of viral replication; intermittent viremia; low viral titer	[17,18,19,20]
HCV	HCV (genotypes 1a, 1b, 4a, 2a, 2c, 3b, 6)	0.03× 10^6^–12.32×10^6^; 6 × 10^5^ –1 × 10^7^ GEs/animal	IP, IV	Adults (6 M; 1 Y)	4–~10^4^ or 10^5^ copies/mL serum	Intrahepatic IFN- β and IL-6 expression changes; anti-NS3 and anti-core Abs produced	Viremia develops; chronic hepatitis; progression of fibrosis; detection of intrahepatic HCV RNA	Individual variations of viral replication; intermittent viremia; low viral titer	[21,22,23]
Influenza A virus	H1N1	10^5^ TCID_50_	IN	3-M-old	10^2.94^–10^4.24^ TCID50/mL	Seroconversion	Infects tree shrews without prior adaptation; replicates in respiratory tracts; develops acute bronchopneumonia and interstitial pneumonia	No loss of appetite, congested eyes and otologic manifestations	[24]
Avian influenza A	H9N2	10^6^ TCID_50_	IN	Adults	10^1.5^–>10^5^ TCID50/mL	Cytokines induced in respiratory tissues; Ab induction	Virus replication in respiratory tissues as observed on histology	Asymptomatic infection; no virus shedding	[25]
Avian influenza A	H5N1	1.0 × 10^6^ PFU	Multisite inoculation	Adult (6M to 4Y-old)	2.8 × 10^3^–5.2 × 10^7^ vRNA copies/mg of lung tissue	IFN-γ, IL-6, TNF-α induced; Ab (IgG) induced at 7 dpi	Replicates in lung tissues; develops severe diffuse pneumonia with fever and weight loss	No sneezing; effect of multisite inoculation on infectivity and pathogenesis yet to be confirmed	[26]
Avian influenza A	H7N9 (A/Anhui/1/2013)	1.0 × 10^6^ PFU	Multisite inoculation	Adult (6M to 4Y-old)	6.5 × 10^1^–3.4 × 10^4^ vRNA copies/mg of lung tissue	IFN-γ induced; Ab (IgG) induced at 7 dpi	Focal pneumonia	No sneezing	[26]
Avian influenza A	H5N1, H7N9, H1N1,	1.0 × 10^6^ EID_50_	IN	1 to 4 M-old	-	Neutralizing Ab induced	Vaccine efficacy testing		[27]
Influenza B virus	Strains of Yamagata or Victoria lineage	1.0 × 10^6^ TCID_50_	IN	Adult	-	Induction of IL-6, IL-10, IP-10, TNF-α and TGF-β mRNA; seroconversion	Weight loss, milder nasal secretions	No significant respiratory symptoms	[28]
SARS-CoV-2	-	10^6^ PFU	IN	Young (6 M to 12 M), adults (2Y to 4 Y) and old (5 Y to 7 Y)	10^5.92^ vRNA copies/mL (nasal wash)	-	Increased body temperature; mild pulmonary abnormalities found in histopathology	No clinical signs develop	[29]
SARS-CoV-2	SARS-CoV-2 strain 107	1 × 10^7^ TCID_50_	Oral, intranasal, ocular	Old (5–6 Y); Adult (1 Y)	1.36 × 10^3^ to 9.32 × 10^5^ copies/g lung tissue	-	vRNA detected in lung tissues and hemorrhage in lung tissues	No viremia; no data on induction of fever	[30]
Human adenovirus- B (HAdV-B)	HAdV-3, HAdV-7, HAdV-14, HAdV-55	5 × 10^5^ TCID_50_	IN	6–8 M old	-	Increased expression of IL-8, IL-10, IL-17A, and IFN-γ; Ab produced	HAdV efficiently replicates and produced high viral load with severe interstitial pneumonia	Presence of pre-existing nAbs against HAdV-3 and HAdV-14	[31]
Dengue virus (DENV)	DENV-2DENV-3	1.5 × 10^3^ PFU	SC, IV	Adult	<2 × 10^4^copies/mL serum	nAb produced	Increased body temperature	Very low viremia, lack of severe dengue manifestations	[16]
Zika virus (ZIKV)	ZIKV strain GZ01	10^6^ PFU	SC	5M	10^4.40^–10^6.02^ RNA copies/mL (1 dpi)	Strong cytokine response induced; nAb produced	Skin rash; suitable for antivirals and vaccine testing	No fever or other neurological and behavioral abnormalities; transient viremia	[32]
Herpes simplex virus type 1 (HSV-1)	HSV-1 strain 17+	1 × 10^6^ PFU	Ocular	6M	-	-	Spontaneous reactivation occurs	All HSV-1 genes are not expressed	[33]
HSV-1	HSV-1 McKrae virus	1 × 10^6^ PFU	Ocular	6M	-	-	Herpes simplex virus keratitis develops; HSV-1 transcripts, ICP0, ICP4, and LAT detected	-	[34,35]
HSV-1, HSV-2	-	25 –10^5^ PFU	SC, IV, IP	Juvenile (1–1.5 M) and adult (5M)	-	-	Higher pathogenicity in juveniles and caused death; herpetic hepatitis; encephalitis and fibrosis in the spleen	-	[36]

TCID_50_, tissue culture dose 50; PFU, plaque-forming units; EID_50_, embryo infectious dose 50; GE, genome equivalents; M, month; Y, year; IN, intranasally; vRNA, viral RNA; dpi, days post-infection; nAb, neutralizing antibody.

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
