# Peer review of "Tree Shrew as an Emerging Small Animal Model for Human Viral Infection: A Recent Overview"

_viruses, 2021, doi:10.3390/v13081641_

Round 1

Reviewer 1 Report

A suitable small animal model is important for the viral pathogenesis and host response study, and then use for evaluation of antiviral drugs and vaccines. Because tree shrews genetically closer to humans than that of rodents, they have the potential to construct viral infection and evaluation model. Until now, tree shrews have been modeled for several important human viral pathogens. This review summarized and commented on tree shrews models of several important human viral pathogens reported, which proved valuable reference for readers in related fields. In this review, the defects and individual variations of each viral infection model were not described enough, which could be supplemented appropriately.

Author Response

Response: We are grateful to the reviewer for his sincere evaluation of the manuscript and suggestion. In line with the reviewer comments, we have added further information as appropriate (line 31-39; 84-85; 173-174; 295-297). 

Reviewer 2 Report

Please see attached file for review.

Author Response

This paper reviews the use of tree shrew as an animal model for the study of various human viral infection. This manuscript comprehensively summarized the application of tree shrew in the study of various human associated virus. I only have several minor suggestions for revision.

Response: We would like to thank the reviewer for his careful evaluation and comments, which will definitely improve the manuscript.

  1. Although in medical community the term “tree shrew” or “Tupaia” refers to Tupaia belangeri (northern tree shrew) or belangeri chinensis (Chinese tree shrew), the tree shrew actually makes up the entire order Scandentia which splits into two families and contains 20 species. In addition, Tupaia is a genus name which includes 16 species. The authors may want to clarify this point.

Response: We would like to thank the reviewer for his comment. Accordingly, we have clarified the statement (line 31-39).

  1. Taxonomically belangeri chinensis is a subspecies of T. belangeri, but they are quite different genetically (Journal of Genetics and Genomics 39: 131-137). In addition, the genetic diversity within T. belangeri is extremely high (Molecular Phylogenetics and Evolution, 60: 358-372). High genetic variation was also noticed within T. belangeri chinensis (Zoological Research 34 (E2): E62−E68). Therefore, caution has to be taken regarding to the source of tree shrew samples.

Response: We are very grateful for the reviewer comments, and we have updated the text accordingly (line 31-39).

  1. Similar to the above, high genetic variation within this species may hinder a wide use of this animal in biomedical research; this might account for the inconsistency of previous results using this animal model. For example, in the studies of HBV and HCV, viral replication varied among individuals. In addition, in consistent results were reported in the studies of SARS-CoV-2. The authors may want to comment on this point.

Response: We are very grateful for the reviewer comments, and in line with reviewer comments we have updated the text (line 84-85; 173-174; 295-296).